# Discovery of Novel Non-Oxime Reactivators Showing In Vivo Antidotal Efficiency for Sarin Poisoned Mice

**DOI:** 10.3390/molecules27031096

**Published:** 2022-02-07

**Authors:** Zhao Wei, Xinlei Zhang, Huifang Nie, Lin Yao, Yanqin Liu, Zhibing Zheng, Qin Ouyang

**Affiliations:** 1Department of Medicinal Chemistry, School of Pharmacy, Air Force Medical University, Xi’an 300071, China; yxxxxjys@fmmu.edu.cn (X.Z.); 13572002997@163.com (H.N.); yaolinpharf@126.com (L.Y.); 2Institute of Pharmacology and Toxicology, Academy of Military Medical Sciences, Beijing 100850, China; lyqf819@aliyun.com; 3Department of Medicinal Chemistry, School of Pharmacy, Third Military Medical University, Chongqing 400038, China

**Keywords:** organophosphate, acetylcholinesterase, non-oximes, reactivators, Mannich phenol

## Abstract

A family of novel efficient non-oxime compounds exhibited promising reactivation efficacy for VX and sarin inhibited human acetylcholinesterase was discovered. It was found that aromatic groups coupled to Mannich phenols and the introduction of imidazole to the ortho position of phenols would dramatically enhance reactivation efficiency. Moreover, the in vivo experiment was conducted, and the results demonstrated that Mannich phenol **L10R1** (30 mg/kg, ip) could afford 100% 48 h survival for mice of 2*LD_50_ sarin exposure, which is promising for the development of non-oxime reactivators with central efficiency.

## 1. Introduction

Organophosphates (**OP**s) including pesticides (e.g., paraoxon, parathion, phorate, dichlorvos and chlorophos, Figure 1) and nerve agents (e.g., sarin, VX, tabun and soman, Figure 1) are highly toxic compounds [1]. **OP**s potently inhibit the cholinergic acetylcholinesterase (**AChE**) through phosphorylation of the enzyme’s catalytic serine residue, and render it incapable of hydrolyzing the neurotransmitter acetylcholine. This inhibition causes accumulation of acetylcholine (**ACh**), and leads to cholinergic crisis, respiratory distress, convulsive seizures and ultimately death [2]. Nerve agents have been used for war and terrorist attacks (e.g., subway attack in Tokyo in 1995) [3]. Organophosphorus pesticides poisoning is also a serious public health issue with about 3,000,000 acute intoxications and over 200,000 fatalities annually worldwide [4].

A combination of intramuscular injections of an **AChE** reactivator of the pyridinium aldoxime family (e.g., pralidoxime (**2-PAM**), trimedoxime (**TMB-4**), obidoxime, **HI-6**, Figure 1) [5,6], a muscarinic receptor antagonist (e.g., atropine), and an anticonvulsant (e.g., diazepam) is approved antidotal therapy for the treatment of **OP** poisoning in humans currently [7,8]. Pyridinium aldoximes are very potent nucleophiles that can break the strong phosphorus oxygen bond of **OP-AChE** and restore the enzyme’s activity [9]. However, due to their permanent positive charge, these quaternary reactivators were poorly distributed in the central nervous system (**CNS**) [10], while the brain was a major target of nerve agents [11]. Consequently, various nonquaternary AChE reactivators were designed and synthesized, such as monoisonitrosoacetone [12,13] and amidine-oximes [14,15], but they were even less efficient than **2-PAM** in vitro. In recent years, a series of pyridyl aldoxime conjugates were reported as superior reactivators for **OP** poisoning in comparison to **HI-6** and obidoxime in vitro [16,17,18,19,20,21], but none of these pyridyl aldoxime conjugates has ever been reported showing in vivo antidotal efficiency. Previously, we had also reported a number of salicylic aldoxime conjugates as efficient nonquaternary reactivators [22,23,24], but they were proven as poor reactivators for sarin poisoned mice in vivo.

It seems that research of nonquaternary oxime reactivators encountered a bottleneck presently. Nevertheless, it was gratifying that two non-oxime intermediates (**L6R1** and **L10R1**, Figure 2) in our previous study were found, showing reactivating ability to sarin and VX inhibited **hAChE** (Figure 3), which represented a totally different reactivator scaffold to the traditional oximes. Moreover, Francine et al. had discovered some similar Mannich phenols (such as **ADQ** and **ADCQ**, Figure 2) exhibiting reactivation efficacy for paraoxon or DFP inhibited **AChE** recently [25,26]; they further found that a series of non-oxime compounds containing imidazole moiety (**SP134** and **SP138,**
Figure 2) displayed reactivating ability [27]. In addition, de Koning et al. found that an imidazole derivative (**3q**, Figure 2) of **ADOC** exhibited reactivating efficacy for **OP** poisoned **AChE** at high concentration (1 mM) [28]. Intriguingly, a series of efficient imidazolium aldoxime reactivators was studied in our previous research [24]. These findings inspired us that maybe imidazole moiety play an important role in the reactivating process of **OP** inhibited **AChE**. As a preliminary structure modification of **L6R1** and **L10R1,** imidazole was introduced to replace the diethylamine moiety in the Mannich phenols. It is gratifying that the resulting compounds (**L6R4** and **L10R4,** Figure 2) exhibited higher in vitro reactivating ability for both sarin and VX inhibited hAChE (Figure 3). Furthermore, a preliminary in vivo experiment disclosed that **L10R1** could afford complete protection for sarin poisoned mice. To the best of our knowledge, **L10R1** was the first reported non-oxime reactivator showing in vivo antidotal efficiency for sarin poisoning.

## 2. Results and Discussion

### 2.1. Synthesis

The synthetic routes to prepare these novel non-oxime compounds were outlined in Figure 1. Firstly, **R1** was obtained through a Mannich reaction by using 4-hydroxy-benzaldehyde **1** and paraformaldehyde in isopropanol. It then underwent a reductive amination reaction with **L6** or **L10** by using Hantzsch ester diludine and iodine to give **L6R1** or **L10R1** in the mixed solvents of dichloromethane and methanol. Synthesis of **L6R4** was commenced with chloromethylation of 4-hydroxybenzaldehyde **1** to give the intermediate **2**; then, condensation of **2** and imidazole in acetonitrile provided **R4**. Finally, reductive amination between **R4** and **L6** afforded compound **L6R4**. **L10R4** was obtained by using a similar reductive amination reaction by using **L10** and **R4.**

### 2.2. In Vitro Inhibition and Reactivation Experiments

The in vitro experiments were conducted with human acetylcholinesterase (**hAChE**) serving as an enzyme source. Two most common nerve agents (VX and sarin) were used for the in vitro reactivation experiment. The enzyme activity was measured using a similar method of Ellman et al. [29]. Firstly, the inhibition experiment was necessary for these novel compounds because strong inhibition of hAChE was unfavorable to the reactivating process of **OP** poisoned enzyme. The in vitro inhibition experiment demonstrated that **L6R1, L6R4,** and **L10R4** were weak inhibitors of **hAChE** with IC_50_ greater than 400 μM, while **L10R1** was a moderate inhibitor (with IC_50_ lower than 40 μM, Table 1). The in vitro reactivation experiment showed that both **L6R4** and **L10R4** exhibited superior reactivation efficacy to **L6R1** and **L10R1** for sarin and VX inhibited **hAChE**, and they even exceeded **HI-6** for VX inhibited **hAChE** at high concentrations (Figure 3). Due to its inhibition potency towards **hAChE**, **L10R1** did not exhibit reactivating ability at high concentrations, but it was confirmed that **L10R1** was able to reactivate poisoned hAChE at low concentrations (Figure 3). 

### 2.3. Determination of Reactivation Kinetics

Determination of maximal reactivation rate constant ***k*_r_**, dissociation constant ***K*_D_** and second order reactivation rate constant ***k*_r_****_2_** (***k*_r_****_2_**
**= *k*_r_****/*K*_D_**) would help obtain a deeper comprehension of the reactivating ability. Results of the reactivation kinetics constants were reported in Table 1. For **L6R1**, introduction of imidazole not only increased reactivation rate constant ***k_r_*** but also enhanced binding affinity towards inhibited **hAChE** (indicated by lower dissociation constant ***K*_D_**), which resulted in dramatically improved reactivation efficacy of **L6R4** in contrast to **L6R1**, especially in the case of VX poisoning. For **L10R4**, introduction of imidazole greatly decreased the inhibition ability of **hAChE**; it was interesting that its inhibition potency to **hAChE** decreased at the same time. Although its reactivation rate constant ***k_r_*** was increased, the decreased binding affinity (higher dissociation constant ***K*_D_**) made **L10R4** a less efficient reactivator. Due to its greatly enhanced binding affinity, **L10R1** was even 2-fold more efficient than **L6R4** and near 3-fold more efficient than **L10R4** for sarin poisoned **hAChE**. Hence, **L10R1** was confirmed as an efficient reactivator for inhibited **hAChE**, but we cannot come to the conclusion that stronger inhibitor would bind tightly to poisoned **hAChE** because, although **L6R1, L6R4,** and **L10R4** were almost equal inhibitors of **hAChE**, their binding affinity for poisoned **hAChE** varied greatly.

It could be concluded that the introduction of imidazole normally increased the reactivation rate constant, which was indicated by higher reactivating ability at relatively high concentrations, but it had the opposite influence on binding affinity for **L10R4**. The p*K*a of these compounds were predicted (Table 1), and it seems that **L6R1** and **L10R1** were a stronger base than **L6R4** and **L10R4**, so they might be easier to protonate under physiological pH, which might increase their binding affinity towards poisoned **hAChE**, but **L10R4** actually did not show binding efficiency. However, it was noteworthy that both **L10R1** and **L10R4** exhibited higher binding affinity to the inhibited **hAChE** than **L6R1** and **L6R4** (see values of ***K*_D_** in Table 1), especially in the case of **L10R1** vs. **L6R1**. Hence, **L10** seemed to be a better ligand than **L6** for the construction of more efficient **AChE** reactivators.

### 2.4. Molecular Docking Simulation

Additionally, we tried to explain the reactivation mechanism of these non-oximes through a molecular docking simulation study (Figure 4). The potential binding pocket was explored according to the ligand from the crystal structure of VX inhibited hAChE in complex with **HI-6** (PDB code: 6CQW, resolution 2.28 Å) [30]. In the case of **L6R1**, the benzamide ring (**L6)** was located at the peripheral anionic site of hAChE, which was fixed via strong hydrophobic and π–π stacking interactions with Trp286 and Tyr72, and it was further stabilized by a H-bond interaction with Val282 and a water molecular, while the Mannich phenol moiety was submerged in the active gorge. In the case of **L6R4**, its ligand **L6** interacted with Trp286 and Tyr 72 in a similar way as **L6R1**, but an H-bond was formed with Glu285 at the peripheral anionic site of hAChE; the molecular was further stabilized by an H-bond between the imidazole moiety and Trp286, which may account for high binding affinity of **L6R4** to VX inhibited hAChE. Ligand **L10** of **L10R1** interact with Trp286 and Tyr 72 through a π–π sandwiching way at the peripheral anionic site, while only weak hydrophobic interactions between **L10** and Tyr 72 existed for **L10R4,** but an additional H-bond between the secondary amine of **L10** and Asp74 was observed during the simulation. The phenolic hydroxyl group of both **L10R1** and **L10R4** formed an H-bond with Phe338 at the active site of hAChE, while **L10R4** was further stabilized by an H-bond interaction between its imidazole moiety and Val282. However, we noticed that the imidazole moiety of **L6R4** and **L10R4** was far from His447, hence we thought that the imidazole moiety might not act as an internal base as histidine mimic in the OP-inhibited triad to promote the reactivation process [25].

For all four of the compounds, we noticed that their nucleophilic phenolic hydroxyl group did not orient to the inhibited Ser203 at the active site of hAChE, so it seemed that these non-oximes did not reactivate inhibited **AChE** through a nucleophilic process as traditional oximes [9]. Given the above analysis, we speculated that interaction between these non-oximes and **OP**-inhibited **AChE** would induce conformation changes of the enzyme, which would help to restart the spontaneous reactivation process in the active gorge. However, the proposed reactivating mechanism was sketchy, and it should be supported by the experimental data. 

### 2.5. In Vivo Biological Experiments

Furthermore, a preliminary in vivo experiment was conducted to test the protection of these new reactivators to mice of sarin exposure. The better reactivators **L10R1, L6R4,** and **L10R4** were selected. In the animal paradigm used herein, mice were observed for neurological toxicity symptoms such as muscles twitching, seizures, and convulsions after sarin or antidotes administration, and 48 h survival was finally recorded. Firstly, animals were pretreated with **L10R1, L6R4**, or **L10R4** at a high dose of 60 mg/kg (ip) to evaluate possible acute toxicity (Table 2, experiment 1). In parallel, two sets of mice pretreated with isotonic saline alone were challenged with 2*LD_50_ dose of sarin (85 μg/Kg), and one set of the mice was treated with atropine sulphate 1 min later (0.5 mg/Kg, control 2 of experiment 1 in Table 2). The results demonstrated that no apparent toxicity was observed for these new reactivators administered alone at a relatively high dose of 60 mg/kg (ip, >180 μM/Kg). For sarin (2*LD_50_ dose) poisoned animals, significant CNS poisoning symptoms were observed (such as muscles twitching, strong seizures, and convulsions) and no saline treated mice (control 1 in experiment 1) survived to the 36 h time point, while only 1/8 of the atropine treated mice (control 2 in experiment 1) survived to the 48 h time point.

Next, in order to maximize these non-oximes’ antidotal ability in a preliminary in vivo experiment, mice were pretreated with different antidotes (including **2-PAM, HI-6, L10R1, L6R4**, and **L10R4**) 15 min before the administration of 2*LD_50_ dose of sarin, and treated with atropine sulfate 1 min later (0.5 mg/Kg). The results demonstrated that the best in vitro non-oxime reactivator **L10R1** exhibited the highest antidotal efficacy in vivo, along with **HI-6** providing complete protection for sarin poisoned mice. More importantly, only slight CNS poisoning symptoms (such as muscles twitching, lack of spontaneous activity, and decreased interest in food consumption) were observed for **L10R1** treated animals, while some mice pretreated with **HI-6** were observed with slight seizures and convulsions. Although the in vitro reactivation efficiency of **L10R1** was inferior to that of **HI-6**, the predicted LogBB and LogP of **L10R1** were much higher than that of **HI-6** (Table 2), which might help **L10R1** to provide higher CNS protection than quaternary **HI-6**, while **HI-6** provided higher peripheral protection and resulted in high survival. However, real data (e.g., **BBB** penetration ability, blood/brain cholinesterase activities) were needed to support these estimations, and our research group would conduct these experiments in the near future. In contrast, **2-PAM** protected only 2/10 mice, the imidazole bearing reactivators **L6R4** and **L10R4** protected only 3/10 and 2/10 mice separately, along with heavy poisoning symptoms such as strong seizures and convulsions, which might be due to their lower reactivation efficiency (***k_r_*_2_**, Table 1) for sarin inhibited **hAChE** and their relatively lower LogBB and LogP (predicted values, Table 2).

Nonetheless, it was noteworthy that **L10R1** exhibited moderate irreversible inhibition ability to **hAChE** and mice were pretreated with **L10R1**, which meant that **L10R1** might serve as a protecting agent for **OP** poisoning in a similar way to pyridostigmine at the same time [31,32]. Given the fact that the weak inhibitors (**L6R4** and **L10R4, Table 1**) of **hAChE** could not provide protection efficiency for sarin poisoned mice, further experiments need to be conducted to understand the complete antidotal mechanism of **L10R1**.

## 3. Experimental Section

### 3.1. Chemicals

All reagents and solvents were used as received from commercial sources. ^1^H NMR and ^13^C NMR spectra were recorded at 400 MHz and 100 MHz on a Bruker-400 instrument in CDCl_3_ or DMSO-d_6_, respectively. Proton and carbon chemical shifts are expressed in parts per million (ppm) relative to internal tetramethylsilane (TMS), and coupling constants (J) are expressed in Hertz (Hz).

### 3.2. Synthesis Procedures for the Preparation of **L6R1**, **L10R1**, **L6R4**, and **L10R4**

**3-((diethylamino)methyl)-4-hydroxybenzaldehyde (R1):** In a 100 mL flask, 4-hydroxybenzaldehyde (**1**, 3.6 g, 29.5 mmol), paraformaldehyde (1.28 g, 42.6 mmol) and diethylamine (3.3 g, 45.1 mmol) were stirred in 25 mL isopropanol, a catalytic amount of concentrated hydrochloric acid (0.3 mL) was added, and the solution was heated to reflux for 2 h. The resulting mixture was cooled to room temperature and purified by silica gel chromatography directly (DCM/MeOH = 20:1, 0.1% NH_4_OH) to afford compound **R1** (4.1 g, 65%) as a pale yellow oil. ^1^H NMR (CDCl_3_, 400 MHz) δ (ppm) 12.22–11.55 (m, 1H), 9.78 (s, 1H), 7.67 (dd, J = 8.3, 1.9 Hz, 1H), 7.53 (d, J = 1.9 Hz, 1H), 6.86 (d, J = 8.3 Hz, 1H), 3.84 (s, 2H), 2.64 (q, J = 7.2 Hz, 4H), and 1.12 (t, J = 7.2 Hz, 6H).

**4-((3-((diethylamino)methyl)-4-hydroxybenzyl)amino)benzamide (L6R1): L6** (0.68 g, 5.0 mmol) and **R1** (1.10 g, 5.4 mmol) were dissolved in a mixed solution of methanol (20 mL) and dichloromethane (20 mL), and diludine (2.53 g, 10.0 mmol) and iodine (1.27 g, 5.0 mmol) were added. The resulting mixture was heated to 50 °C and stirred for 4 h. After concentration under reduced pressure, the residue was purified by silica gel chromatography (DCM/MeOH = 20:1, 0.1%NH_4_OH) to afford the compound **L6R1** (1.2 g, 80%) as a white solid. ^1^H NMR (400 MHz, DMSO) δ 10.83–8.06 (m, 2H), 7.61 (d, *J* = 8.2 Hz, 2H), 7.55 (s, 1H), 7.35 (s, 1H), 7.27 (d, *J* = 7.8 Hz, 1H), 6.91 (d, *J* = 8.2 Hz, 2H), 6.71 (s, 1H), 6.56 (d, *J* = 8.1 Hz, 2H), 4.23 (d, *J* = 4.3 Hz, 2H), 4.18 (s, 2H), 3.05 (d, *J* = 6.8 Hz, 4H), 1.21 (t, *J* = 6.7 Hz, 6H).^1^H NMR (400 MHz, DMSO-D_2_O) δ 7.62 (d, *J* = 8.4 Hz, 2H), 7.34 (s, 1H), 7.29 (d, *J* = 8.2 Hz, 1H), 6.92 (d, *J* = 8.3 Hz, 1H), 6.58 (d, *J* = 8.4 Hz, 2H), 4.24 (s, 2H), 4.17 (s, 2H), 3.04 (d, *J* = 7.1 Hz, 4H), 1.22 (t, *J* = 7.0 Hz, 6H). ^13^C NMR (101 MHz, DMSO) δ 168.36, 155.02, 151.02, 131.37, 130.29, 130.01, 128.95(2*C), 120.60, 116.24, 115.36, 111.20(2*C), 50.37, 46.58(2*C), 45.16, 8.46(2*C). HRMS (ESI^+^) *m/z* calcd for C_19_H_26_N_3_O_2_^+^ 328.2025 found 328.2020 Da (Appendix A).

**4-(((1H-pyrazolo[3,4-b]pyridin-3-yl)amino)methyl)-2-((diethylamino)methyl)phenol (L10R1): L****10** (0.26 g, 1.9 mmol) and **R1** (0.43 g, 2.1 mmol) was dissolved in a mixed solution of methanol (8 mL) and dichloromethane (8 mL), and diludine (1.01 g, 4.0 mmol) and iodine (0.51 g, 2.0 mmol) were added. The resulting mixture was heated to 50 °C and stirred for 6 h. After concentration under reduced pressure, the residue was purified by silica gel chromatography (DCM/MeOH = 15:1, 0.1%NH_4_OH) to afford the compound **L****10R1** (0.36 g, 57%). ^1^H NMR (400 MHz, DMSO) δ 11.94 (s, 1H), 8.34 (d, *J* = 3.3 Hz, 1H), 8.17 (d, *J* = 7.7 Hz, 1H), 7.14 (d, *J* = 8.2 Hz, 1H), 7.11 (s, 1H), 6.94 (dd, *J* = 7.8, 4.6 Hz, 1H), 6.65 (d, *J* = 8.1 Hz, 1H), 6.59 (t, *J* = 5.6 Hz, 1H), 4.33 (d, *J* = 5.5 Hz, 2H), 3.71 (s, 2H), 2.67–2.52 (m, 4H), 1.02 (t, *J* = 7.1 Hz, 6H). ^1^H NMR (400 MHz, DMSO-D_2_O) δ 8.42–8.31 (m, 1H), 8.18 (d, *J* = 7.9 Hz, 1H), 7.19 (d, *J* = 8.2 Hz, 1H), 7.16 (s, 1H), 6.99 (dd, *J* = 7.9, 4.6 Hz, 1H), 6.70 (d, *J* = 8.1 Hz, 1H), 4.35 (s, 2H), 3.78 (s, 2H), 2.62 (q, *J* = 7.0 Hz, 4H), 1.05 (t, *J* = 7.1 Hz, 6H).^13^C NMR (101 MHz, DMSO) δ 155.90, 152.22, 148.80, 148.57, 130.46, 129.74, 128.70, 127.95, 121.16, 115.00, 113.97, 105.91, 54.47, 45.79 (2*C), 45.63, 10.50 (2*C). HRMS (ESI^+^) *m/z* calcd for C_18_H_24_N_5_O^+^ 326.1981 found 326.1975 Da (Appendix A).

**3-(chloromethyl)-4-hydroxybenzaldehyde (2)**: 4-hydroxybenzaldehyde (12.38 g, 101 mmol) and paraformaldehyde (3.3 g, 110 mmol) were added to 100 mL concentrated hydrochloric acid in a 250 mL two-neck round flask; the mixture was stirred at 65 °C for 2 h and then cooled to room temperature. The resulting mixture was extracted with ethyl acetate (EA, 2 × 280 mL), and the combined extracts were dried over anhydrous Na_2_SO_4_ and concentrated under vacuum to give a crude product, which was purified by recrystallization from mixed solution of EA and PE to afford compound **2** (6.2 g, 36%) as a white solid. ^1^H NMR (DMSO 400 MHz) δ (ppm) 11.16 (s, 1H), 9.81 (s, 1H), 7.91 (d, *J* = 1.9 Hz, 1H), 7.77 (dd, *J* = 8.4, 1.9 Hz, 1H), 7.04 (d, *J* = 8.4 Hz, 1H), 4.76 (s, 2H).

**3-((1H-imidazol-1-yl)methyl)-4-hydroxybenzaldehyde (R4)**: A mixture of imidazole (0.45 g, 6.6 mmol), N,N-Diisopropylethylamine (DIEPA_,_0.85 g, 6.6 mmol), and tetrabutylammonium bromide (TABA, 0.43g, 1.3 mmol) in acetonitrile (20 mL) was stirred at room temperature, **2** (1.0 g, 6.4 mmol) was added, and the reaction mixture was stirred for 12 h. After concentration under reduced pressure, the residue was purified by silica gel chromatography (DCM/MeOH = 15:1, 0.1%NH_4_OH) to afford the compound **R4** (0.4 g, 45%) as a white powder. ^1^H NMR (DMSO, 400 MHz) δ (ppm) δ 11.42–11.08 (m,1H), 9.75 (s, 1H), 7.82–7.67 (m, 2H), 7.54 (d, *J* = 1.6 Hz, 1H), 7.19 (s, 1H), 7.03 (d, *J* = 8.3 Hz, 1H), 6.91 (s,1H), 5.18 (s, 2H) (Appendix A).

**4-((3-((1H-imidazol-1-yl)methyl)-4-hydroxybenzyl)amino)benzamide (L6R4): L6** (0.35 g, 2.8 mmol) and **R4** (0.55 g, 2.7 mmol) was dissolved in a mixed solution of methanol (20 mL) and dichloromethane (20 mL), diludine (1.39 g, 5.5 mmol); 5 Å molecular sieves (0.28 g) and iodine (0.64 g, 2.5 mmol) were added. The resulting mixture was heated to 50 °C and stirred for 8 h. After concentration under reduced pressure, the residue was purified by silica gel chromatography (DCM/MeOH = 15:1, 0.1%NH_4_OH) to afford the compound **L6R4** (0.36 g, 41%) as a white solid. ^1^H NMR (DMSO, 400 MHz) δ (ppm) δ 14.23 (s, 1H), 10.03 (s, 1H), 9.15 (s, 1H), 7.75–7.46 (m, 4H), 7.30 (s, 1H), 7.21 (s, 1H), 6.97–6.72(m, 2H), 6.63–6.45 (m, 2H), 5.31 (s, 2H), 4.19 (s, 2H). ^1^H NMR (DMSO-D_2_O, 400 MHz) δ (ppm) δ 9.06 (s, 1H), 7.72–7.45 (m, 4H), 7.30 (s, 1H), 7.20 (s, 1H), 6.87–6.76 (m, 1H), 6.74–6.66 (m 1H), 6.45 (m, 2H), 5.29 (s, 2H), 4.19 (s, 2H).^13^C NMR (CDCl_3_, 100 MHz) δ = 154.37, 152.68, 149.33, 148.98, 136.75, 131.50, 130.21(2*C), 129.72, 124.82, 122.27, 121.10, 115.48, 114.52(2*C), 106.37, 48.94, 46.78. HRMS (ESI^+^) *m/z* calcd for C_18_H_19_N_4_O_2_^+^ 323.1508 found 323.1503 Da (Appendix A).

**2-((1H-imidazol-1-yl)methyl)-4-(((1H-pyrazolo[3,4-b]pyridin-3-yl)amino)methyl)phenol (L10R4): L****10** (0.13 g, 0.97 mmol) and **R4** (0.19 g, 0.95 mmol) were dissolved in a mixed solution of methanol (8 mL) and dichloromethane (8 mL); diludine (0.48 g, 1.9 mmol), 5 Å molecular sieves (0.15 g), and iodine (0.23 g, 0.9 mmol) were added. The resulting mixture was heated to 50 °C and stirred for 10 h. After concentration under reduced pressure, the residue was purified by silica gel chromatography (DCM/MeOH = 12:1, 0.1%NH_4_OH) to afford the compound **L6R4** (0.11 g, 36%). ^1^H NMR (DMSO, 400 MHz) δ (ppm) δ 14.23 (s, 2H), 10.06 (s, 2H), 9.16 (s, 2H), 8.63-8.28 (m, 2H), 7.81–7.54 (m, 2H), 7.39 (s, 1H), 7.30 (s, 1H), 7.15–6.96 (m, 1H), 6.93–6.72 (m, 1H), 5.34 (s, 2H), 4.40 (s, 2H).^1^H NMR (DMSO-D_2_O, 400 MHz) δ (ppm) δ 9.07 (s, 1H), 8.61-8.29 (m, 2), 7.69–7.51 (m, 2H), 7.38 (s, 2H), 7.28 (s, 2H), 7.14–6.97 (m, 1H), 6.93–6.73 (m, 1H), 5.31 (s, 2H), 4.38 (s, 2H).^13^C NMR (DMSO, 100 MHz) δ = 155.04, 149.35, 148.30, 147.16, 137.80, 135.60, 130.69, 130.50, 130.12, 122.45, 120.95, 120.14, 115.73, 113.75, 108.31, 48.36, 46.18. HRMS (ESI^+^) *m/z* calcd for C_17_H_17_N_6_O^+^ 321.1464 found 321.1458 Da (Appendix A).

### 3.3. Computational Methods

Molecular docking simulations were conducted by using the “SYBYL-X 2.0” software. The potential binding pocket was explored according to the ligand from the crystal structure of VX inhibited hAChE in complex with HI-6 (PDB code: 6CQW, resolution 2.28 Å) and from the crystal structure of hAChE in complex with HI-6 (PDB code: 6CQU, resolution 2.308 Å) [7]. The main protocols and the parameters set for the docking were as follows: (1) Additional starting conformations per molecule were set to 10. (2) Max number of rotatable bonds per molecule was set to 100. (3) Maximal number poses per molecule were set to 20. (4) Density of search and number of spins per alignment were set to 9.0 and 20, respectively. (5) Pre-dock minimization, post-dock minimization, molecule fragmentation, ring flexibility, and soft grid treatment were turned on in the present work.

### 3.4. General In Vitro AChE Screening Information

Human acetylcholinesterase (**hAChE**, 20 U/mL, dissolved in 20 mM HEPES, pH 8.0), bovine serum albumin (BSA), acetylthiocholine (ATCh), and 5, 5-dithiobis-2-nitrobenzoic acid (DTNB) were purchased from Sigma-Aldrich. **HI-6** and obidoxime were synthesized according to the literature protocols [33,34]. Sarin and VX were from the Anti Chemical Command and Engineering Institute of the Chinese People’s Liberation Army. (Caution! **OP**s used in our research are highly toxic and must be handled with extreme care by well-trained personnel. Use of these materials has been approved by the Anti Chemical Command and Engineering Institute of the Chinese People’s Liberation Army. After reactivation studies, biochemical samples were neutralized by stirring with 2 M NaOH for 12 h. and the remaining solutions were brought back to pH~7 and disposed in chemical waste.) In addition, 10 mM concentration solutions of the final compounds were prepared in water containing 20% methanol. They were further diluted by PBS (0.1 M, pH 7.4) to 3 mM concentrations. It was found that there was no effect of methanol on hAChE by a control experiment. All the biological evaluation experiments were conducted in a 96-well plate; the enzyme activity was measured by the time-dependent hydrolysis of ATCh in which the product (thiocholine) was detected by reaction with the Ellman’s reagent, 5, 5’-dithiodis-2-nitrobenzoic acid (DTNB), and absorbance at 412 nm [29].

### 3.5. Procedures of hAChE Inhibition Experiments

The procedures of inhibition experiments were as follows:(1)A stock solution of hAChE was diluted 2000-fold with PBS (0.1 M, pH = 7.4, 0.1% BSA);(2)To 20 μL of the diluted enzyme, 10 μL reactivator solutions (reactivator final concentrations: 10, 50, 200, 500, and 1000 μM, and each sample was measured in duplicate in parallel in a 96-well plate) were added, and the mixture was incubated for 15 min at 25 °C. A positive control was run in parallel by adding 10 μL of PBS instead of reactivator solution to the enzyme.(3)For each sample in 96-well plate, 30 μL of ATCh (3.0 mM, pH = 7.4 PBS), and 150 μL of DTNB (0.75 mM, pH = 7.0 PBS) were added. Then, the resulting mixture was centrifuged at 4 °C for 1 min to remove bubbles, and the reaction product was monitored immediately by testing the absorption value at 412 nm (0 < abs < 3).

Enzyme activity was calculated by using the formula: **%Inhibition = 100 − 100*S/P**, where **S** = absorption value of the tested sample, and **P** = absorption value of the positive control. IC_50_ values were calculated by nonlinear fitting using the standard IC_50_ equation: **%Inhibition = 100 − 100*IC_50_/(IC_50_ + [****R])**. [**R**] = concentrations of the reactivators.

### 3.6. Procedures of Reactivation Experiments

The procedures of reactivation experiments were as follows:(1)A stock solution of hAChE was diluted 2000-fold with PBS (0.1 M, pH = 7.4, 0.1% BSA); the concentrations of different nerve agents were determined by a pre-experiment similar to the inhibition experiment to attain an inhibition plateau from 90% to 95%. We tried carefully to control the dosage of **OP** used to avoid 100% inhibition of hAChE, which meant that all **OP** used had bound to the enzyme, and there was no **OP** presented in the reaction mixture. The final concentrations of **OP**s were as follows: VX, 3*10^7^ fold diluted; sarin, 1.6*10^6^ fold diluted.(2)The diluted hAChE (20 μL) was incubated with different nerve agents (10 μL) at 25 °C for 15 min. Then, the inhibited enzyme was incubated with reactivators (15 μL, 300/150/75/30 μM) at 25 °C for 30 min (final concentrations of reactivators were 100/50/25/10 μM).(3)For each sample in a 96-well plate, 30 μL of ATCh (3.0 mM, pH = 7.4 PBS) and 150 μL of DTNB (0.75 mM, pH = 7.0 PBS) were added. Then, the resulting mixture was centrifuged at 4 °C for 1 min to remove bubbles, and the reaction product was monitored immediately by testing the absorption value at 412 nm (0 < abs < 2). Blank samples were run in parallel and consisted of: **(a)** a positive control **(P)**: an uninhibited enzyme (20 μL) was used instead of the inhibited enzyme; **(b)** a negative control **(N)**: PBS (25 μL, 0.1 M, pH 7.4) was used instead of reactivators. %Reactivation was calculated using the formula: **%Reactivation = 100*(S-N)/(P-N)**.

### 3.7. Determination of Reactivation Kinetics

To further investigate the reactivating mechanism, the %reactivation at different time intervals and at different concentrations were measured by using the same method we described in Section 3.6. The observed first-order rate constant ***k_obs_*** for each reactivator concentration, the dissociation constant ***K_D_*** of inhibited enzyme–reactivator complex (**EP–****R**), and the reactivation rate constant ***k_r_*** were calculated by nonlinear fitting using the standard reactivator concentration dependent reactivation equation derived from the following scheme [35,36]:



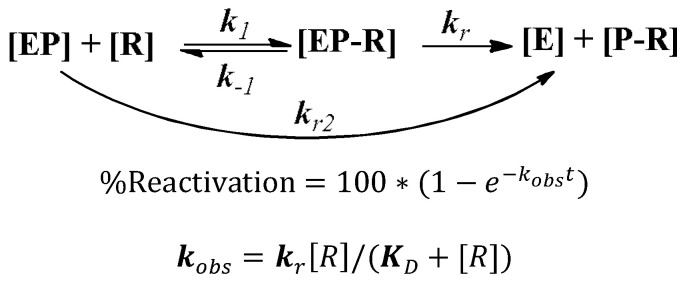



In this scheme, **EP** is the phosphylated enzyme, **[EP–****R]** is the reversible Michaelis-type complex between **EP** and the reactivators **[****R]**, **E** is the active enzyme and **P–****R** the phosphylated reactivator or hydrolysised **OPs**. ***K_D_*** is equal to the ratio **(*k***
_−***1***_
**+ *k_r_*)/*k_1_***, and it typically approximates the dissociation constant of the **[EP–****R]** complex, where it follows that: ***k_r2_* = *k_r_*/*K_D_***.

For more details, we conducted the reactivation experiment at different reactivator concentrations with discontinuous determination of enzyme activity at different reactivation times. Basically, the concentration of the reactivated AChE is proportional to the enzyme activity, ***k****_obs_*** was calculated from the continuous recording of d[S]/dt, and the velocity of substrate hydrolysis (*v*) may be expressed as a pseudo-first-order process of reactivation Equation (1):(1)$$\ln\left(\frac{v_{0}-v_{t}}{v_{0}-v_{i}}\right)=-k_{obs}t$$
in which *v_t_* represents velocity at time *t*, *v*_0_ represents maximum velocity (normal control), and *v_i_* represents minimum velocity (poisoned control). Alternatively, for each reactivator concentration, the ***k_obs_*** value was determined by linear regression analysis applying Equation (2):(2)$$v_{t}=v_{0}\left(1-e^{-k_{obs}t}\right)$$

Integration of (2) results in Equation (3):(3)$$-d\left[S\right]=\int_{0}^{t}vdt=v_{o}t+\frac{v_{o}}{k_{obs}}\left(e^{-k_{obs}t}-1\right)$$
which was used for nonlinear regression analysis of the data points from individual reactivator concentrations.

Concentrations of the reactivators used to determine the concentration dependence of the apparent reactivation rate ***k_obs_*** for the reactivation of OPs inhibited *h*AChE were shown in Table 3, plots of ***k_obs_*** vs. concentrations of **HI-6** and the new synthesized compounds were shown in supporting information **[37]**. Due to different effect-acting concentrations for different reactivators, different concentration scales were used for **HI-6** and **L10R4** or other phenols.

### 3.8. Details of the In Vivo Reactivation Experiments

In order to maximize the antidotal ability of these new reactivators, an atropine dose as low as possible should be used. For the in vivo experiment, if only atropine was used for sarin (2*LD_50_ dose) poisoned animals, we found that a dose of 0.5 mg/Kg would result in 1/8 of mice (control 2 in experiment 1) surviving to the 48 h time point. The procedure of in vivo protection experiments for sarin exposure was as follows:

For experiment 1:Animals were pretreated with **L10R1, L6R4**, and **L10R4** at a dose of 60 mg/kg (ip); in parallel, two sets of mice pretreated with isotonic saline alone were challenged with 2*LD_50_ dose of sarin (85 μg/Kg), and one set of the mice was treated 1 min later with atropine sulfate (0.5 mg/Kg, control 2 in Table 2).Mice were observed for neurological toxicity symptoms such as muscles twitching, seizures, and convulsions after sarin or antidote administration, and the 48 h survival was finally recorded.

For experiment 2:Mice were pretreated with different antidotes (including **2-PAM, HI-6, L10R1, L6R4,** and **L10R4**);15 min later, a 2*LD_50_ dose of sarin (85 μg/Kg) was administrated (ip), and atropine sulfate (0.5 mg/Kg) was administrated 1 min later.Mice were observed for neurological toxicity symptoms such as muscles twitching, seizures, and convulsions after sarin or antidote administration, and the 48 h survival was finally recorded.

## 4. Conclusions

In conclusion, a family of novel non-oxime compounds displayed promising reactivation efficacy for **VX** and sarin inhibited **hAChE** were discovered in this paper. **L6R4, L10R1**, and **L10R4** were proven as efficient reactivators for sarin and **VX** inhibited **hAChE** in vitro. Aromatic groups coupled to Mannich phenol seemed to be key structures for construction of efficient reactivators, while the introduction of imidazole to the *ortho* position of phenols would promote the reactivating ability at high concentrations, but it decreased binding affinity towards the poisoned **hAChE** and the resulting **L6R4** did not exhibit superior reactivating ability to **L10R1** at low concentrations. Moreover, due to its improved in vitro reactivating efficiency and lipophilicity, **L10R1** emerged as a potential and efficient antidote which afforded complete 48 h protection in an animal survival experiment of 2*LD_50_ dose sarin exposure. Different from the traditional quaternary oxime reactivators, a totally novel nonquaternary non-oxime structural scaffold was explored, and exciting reactivation results were obtained in this study. These findings provided a completely new starting point for the development of improved reactivators with centrally antidotal efficiency.

## Data Availability

Not applicable.

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
