# Peer review of "Discovery of Novel Non-Oxime Reactivators Showing In Vivo Antidotal Efficiency for Sarin Poisoned Mice"

_molecules, 2022, doi:10.3390/molecules27031096_

Round 1
Reviewer 1 Report
Although pyridinium aldoximes are efficient nucleophiles that could break down the nerve agent, their permanent positive charge prevented them to pass through BBB. Other alternative structures were developed by the science community; however, those structures are not as effective as the pyridinium aldoximes. The manuscript entitled “Discovery of novel non-oxime reactivators showing in vivo antidotal efficiency for sarin poisoned mice” by Zhao Weil and co-workers try to resolve this challenge by designing and synthesizing 4 non-oxime analogs. The following questions should be addressed in order to publish in Molecules.
1. There are only 4 compounds synthesized in this submission, its difficult to discuss the meaningful SAR only with 4 compounds. The authors should explain why these 4 compounds were chosen in the first place.
2. In the in vitro experiments, the authors chose to use HI-6 as the control; however, 2-PAM was also included as the control in the in-vivo study, is there any justification to it?
3. The authors published their work on 2018 by using 3 compounds (2-PAM, HI-6, and obidoxime) as the control; however, they only use 2 in this report. Is there any reason for it?
4. In line 300, the chemical shift was reported as: 6.666.45, which I believe was a mistake and should be corrected.
5. The docking model described in this submission does not really explain the re-activating property of the synthesized compound. I would suggest the authors to remove this section.
6. The writing of this manuscript requires minor English editing.
I would recommend accepting this report after the authors address the above-mentioned questions.
Reviewer 2 Report
Dear Authors, firstly I would like to congratulate for the great work, as much research is needed to tackle the limitation of current pyridinium oximes available to treat organophosphorus poisoning. I understand your study as much important and shed light to many others in the field. All the sections were well written, the experiments and the "idea behind" described in clear manner. I have just few comments given below, and I hope you can address them for further publication. Best regards and the best of luck in further studies (especially on the discussion of the reactivation mechanism of your compounds).
- Page 1, Line 21: In the Figure 1, the structure of phorate was not presented.
- Page 3, Line 77: I would suggest including diludine after Hantzsch ester, to keep the information as clear as possible. Although I understand that for a person skilled in the art may appear obvious, I walways like to think about to “non-synthetic” readers.
- Page 3, Scheme 1: Imidazole is missing in the conditions given for step d). Please confirm.
- Page 8, Line 284: I think Authors mean diethylamine instead of DIEPA. Please confirm.
- In all synthetic procedures, when informing the purification conditions, eluent is written as “(DCM/MeOH = 20:1, 0.1% NH3H2O)”. Could you please confirm if “NH3H2O” is correct? I would suggest informing as “NH4OH”, which I understand as an additive to the mobile phase. Is this an isocratic elution condition? Some procedures do not present work up conditions (example: synthesis of R1, concentration, filtration), whereas others present. I suggest you informing the conditions, although I understand that for a person skilled in the art may appear obvious.
- For novel compounds, I understand that the NMR spectra must be available in the supplementary material.
